# Over 25 Years of Pediatric Botulinum Toxin Treatments: What Have We Learned from Injection Techniques, Doses, Dilutions, and Recovery of Repeated Injections?

**DOI:** 10.3390/toxins12070440

**Published:** 2020-07-06

**Authors:** Heli Sätilä

**Affiliations:** Department of Neuropediatrics, Päijät-Häme Central Hospital, Keskussairaalankatu 7, 15850 Lahti, Finland; heli.satila@phhyky.fi; Tel.: +358-44-719-5238

**Keywords:** botulinum toxin type A, cerebral palsy, injection techniques, dosage, dilution, motor endplate targeted injections, muscle atrophy, repeated injections

## Abstract

Botulinum toxin type A (BTXA) has been used for over 25 years in the management of pediatric lower and upper limb hypertonia, with the first reports in 1993. The most common indication is the injection of the triceps surae muscle for the correction of spastic equinus gait in children with cerebral palsy. The upper limb injection goals include improvements in function, better positioning of the arm, and facilitating the ease of care. Neurotoxin type A is the most widely used serotype in the pediatric population. After being injected into muscle, the release of acetylcholine at cholinergic nerve endings is blocked, and a temporary denervation and atrophy ensues. Targeting the correct muscle close to the neuromuscular junctions is considered essential and localization techniques have developed over time. However, each technique has its own limitations. The role of BTXA is flexible, but limited by the temporary mode of action as a focal spasticity treatment and the restrictions on the total dose deliverable per visit. As a mode of treatment, repeated BTXA injections are needed. This literature reviewed BTXA injection techniques, doses and dilutions, the recovery of muscles and the impact of repeated injections, with a focus on the pediatric population. Suggestions for future studies are also discussed.

## 1. Introduction

Botulinum toxin type A (BTXA) has been used in the treatment of pediatric lower and upper limb hypertonia for over 25 years, mostly in cerebral palsy (CP). As younger children appear to have more dynamic spasticity and less fixed contractures than older patients, they may respond better to BTXA treatment [1,2]. In the lower limb, BTXA is used for facilitating milestones, improving function and posture, easing pain or care, and as a means of “buying time” until the child is sufficiently mature for more definitive procedures [1,3]. Spastic equinus management with BTXA is regarded as useful from age two to six years but in most children over the age of six years requires muscle-tendon lengthening [4]. This does not, however, preclude the use of BTXA in selective muscles following surgical procedures.

Later treatment may still be beneficial in terms of pain relief, ease of care, and posture, such as standing, sitting, and reaching [1,3]. Children with spastic upper limb deformities, with moderate muscle tone (grade 2 or 3 on the Modified Ashworth Scale; MAS), has the ability for voluntary movements in the wrist or fingers, and with good motivation and capacity for motor learning, benefit the most from BTXA treatment [5,6]. Adequate pre-injection grip strength has been found to correlate with a positive functional response [7]. BTXA is administered as a single or multilevel treatment. The objectives are to “break” the abnormal strict movement pattern by injecting the spastic muscles, and allow the training and strengthening of both, the treated and antagonistic muscles, and to enable overall flexibility and better alignment [6,8,9]. BTXA was helpful for pre-operative decision-making in the upper and lower limb surgery [1,10].

The toxin is produced by the anaerobic spore-forming bacteria *Clostridium botulinum* of which seven immunologically distinct serotypes designated with alphabetical letters from A to G have been identified. They exert their effect by blocking the release of the neurotransmitter acetylcholine at cholinergic nerve endings of the skeletal and autonomic nervous system. A selective and temporary chemical denervation ensues, causing clinically detectable muscle weakness and atrophy. Neurotoxin types A (OnabotulinumtoxinA/Botox, AbobotulinumtoxinA/Dysport, IncobotulinumtoxinA/Xeomin) and B (RimabotulinumtoxinB/Neurobloc/Myobloc) have been introduced into clinical practice, and type A is the most widely used serotype in the neuropediatric population.

The adverse events reported have usually been transient and mild, such as focal weakness, pain at the injection site, bruising, tripping, a local rash, and influenza-like illness. Fewer reports have included symptoms, such as urinary or fecal incontinence, generalized weakness, worsening of strabismus or dysphagia, irritability, or constipation. Few deaths have been reported in the literature, and caution is recommended in children with pre-existing bulbar symptoms, gastro-esophageal reflux, or frequent chest infections, as these conditions expose patients to aspiration pneumonia [3]. Other contraindications include myasthenia gravis and the concomitant use of amino glycoside antibiotics or non-depolarizing muscle relaxants.

General recommendations and consensus statements on the clinical use and injection techniques of BTXA have been published [3,11,12,13,14]. Accurate injection, directed as close as possible to the neuromuscular junctions (NMJ), is considered to be a prerequisite for efficient treatment [15,16,17]. As more information on the sites of NMJs in upper and lower extremity muscles has become available, the exact targeting of injections is now possible. The role of BTXA is flexible but finite due to the short time of action and restrictions on the total dose deliverable per visit [2,16]. Thus, repeated BTXA injections are needed.

In this review article, the current literature on injection techniques, doses and dilutions, the recovery of muscles from BTXA injections, and the implications for treatments from a pediatric point of view are highlighted. The information was collected from MEDLINE and PUBMED with the keywords “Botulinum toxin type A” and “Injection techniques”/ “Multiple site injections”/“Localization techniques”/“Motor endplate targeted injections”/“Dosage”/“Dilution”/“Diffusion”/“Muscle effects”/“Muscle atrophy”/“Recovery”/“Repeated injections”/“Repeated treatment” until April 2020. The focus was on pediatric studies with patients under the age of 18 but relevant animal and adult studies were included as well.

## 2. Site of Action: The Motor Endplate Zone

BTXA molecules are synthesized as single polypeptide chains that are only weakly toxic. Either in the host bacterium or at the final destination, the molecule undergoes two major changes, nicking and disulfide bond reduction, both of which increase the potency of the toxin [18]. The heavy chain carboxyl end binds the toxin molecule specifically to cholinergic neurons in the NMJs and the amino acid end is important for translocation of the light chain from the endocytosed vesicle into the cytosol (Figure 1).

Once in the neuron cytosol, the disulfide bond is reduced, and the toxin is activated by converting the light chain into a proteolytic enzyme. From the site of adsorption (intestine, wounds, or injection into the muscle), BTXA diffuses and binds with high affinity to the peripheral cholinergic nerve endings in the preganglionic sympathetic and parasympathetic nervous system, the postganglionic parasympathetic nervous system, and the efferent motor nerves at the NMJ [18]. The presynaptic blocking of the release of acetylcholine (Ach) involves five stages: Binding, internalization, translocation, action in the cytosol, and cleavage of synaptosome-associated protein-25 (SNAP-25), one of the proteins forming the SNARE (soluble N-ethyl-maleimide-sensitive factor attachment protein receptor) complex, which is essential for Ach-release [18]. A temporary selective denervation ensues, causing clinically detectable skeletal muscle weakness and atrophy.

The time course of the effects of BTXA assessed by serial electroneuromyography (EMG) recordings in healthy adult extensor digitorum brevis (EDB) muscle showed that the amplitude of the muscle action potential began to decline 48 h after the injection, peaked between 1 and 3 weeks and faded gradually. The entire recovery process required 2–4 months [19]. The degree and, to some extent, the duration of the effect were dose-dependent [20]. The BTXA action on the autonomic nervous system was not found to differ from its action on the striate muscle but lasted longer (1 year) [21].

In mammalians and humans, the motor endplates (MEP) at the NMJs were found to situate at the midpoint of each muscle fiber, forming a MEP zone [22,23]. The form of this zone depends on the fiber length and muscle configuration and should be regular in a given muscle, but differs in different muscles [22,23,24]. The MEPs may either be scattered or constitute one or multiple zone-like patterns [23]. Therefore, if the muscle fiber architecture is known, the site of the MEP zone is predictable. Attempts have been made by various methods to locate and map the MEP bands in different human muscles, both in the upper and lower limbs [24,25,26,27,28,29,30,31,32,33,34]. These works enable a clinician working with children to extrapolate the maps to patients of varying sizes. An example of the MEP zones in the gastrocnemius muscle is depicted in Figure 2.

## 3. Site of Injections: Near the Motor Endplate Zone

In theory, injections near the MEP zone would enhance the efficacy and potentially reduce the required doses, immunoresistance, side-effects, and costs. There is evidence from both animal and human studies that the injection distance to MEP zones influences the effect of BTXA treatment [35,36,37,38,39,40,41,42,43]. In rats, tibialis anterior (TA) muscle BTXA injections given 0.5 cm from the MEP zone yielded 50% less paralysis, and no paralysis was observed 1 cm from the MEPs [35]. The animals were killed after 24 h so more paralysis could have occurred. Likewise, it is not known how well this indirect histochemical model parallels with functional paralysis.

A pediatric study tested proximal (near motor point) and distal (mid-belly, near the MEPs) injection points in gastrocnemius (GC) muscles in children with CP [37]. A single-point injection of 3 “mouse units” (mu or U)/kg of onabotulinumtoxinA per site was used. No difference was noted in the range of motion (ROM), dynamic muscle length (with Tardieu scale), calf tone (with MAS), or video gait analysis. However, the median change from the baseline in the video gait analysis showed significant differences favoring the distal group at two months, suggesting that the injections close to the MEP area acted more promptly.

Two other studies on children with CP used MEP-targeted onabotulinumtoxinA injections into the iliopsoas and gracilis muscles showing that the effect on the outcomes studied (muscle atrophy by magnetic resonance imaging (MRI) and reduced muscle activity by EMG) was better when compared with injections distant from the MEPs [38,39]. In these studies, function was not investigated. In adult studies with stroke patients comparing GC single-point proximal and distant injection sites, no differences between the groups in surface EMG, muscle tone, ankle ROM, dynamic muscle length, or gait were detected [40,41]. It may be that the outcome measures used in the human studies were not sensitive enough to detect the differences, or that the injection technique used (manual palpation, MP) affected the results.

One possibility is that the autofocusing high tropism property of BTXA enables the toxin to reach the MEP zones effectively. In a four-month follow-up double-blind study with hemiplegic adults where the patients were randomized into three groups (receiving onabotulinumtoxinA 100 U/mL dilution (0.4 mL/site) into four quadrants, or receiving 100 U/mL dilution (0.4 mL/site) into four sites along the MEP zone, or injected with 20 U/mL dilution (2 mL/site) into four quadrants), a low-volume BTXA near the MEP zone or a high-volume dilution distant from the MEP zone was more effective than the low-volume injected distantly [42]. In healthy adult EDB muscle, a 46% reduction in effectiveness was found if BTXA was injected over 1 cm from the MEP zone [43].

The use of single or multiple injection sites stemmed from the practicality of dividing the larger dose and volume into multiple sites in a specific muscle to avoid causing unwanted spreading and adverse effects [3,13,14,16]. The number of injection sites may be determined by the MEP zone configuration or the size of the muscle. In little muscles (e.g., in the hand or forearm), BTXA presumably suffuses well through the whole muscle. The risk of spread into adjacent muscles increases with higher doses and volumes. In big muscles (e.g., in lower extremities or biceps brachii in the upper extremities), higher doses may have to be used, and dividing the dose along the presumed MEP zone might increase the effect. It may be appropriate to divide the toxin even when injecting muscles with one MEP zone.

Studies exploring single versus multiple injection sites are few, with some reports showing better effects with multiple site injections and some reporting no difference between the two techniques [40,42,44,45,46]. One small study with 17 children comparing single and multisite injections of the GC muscle found no differences in the ROM, dynamic muscle length, tone, gait pattern, or attained goals in the Goal Attainment Scale (GAS) between the groups [47]. The total number of muscles needing treatment, or the availability of general anesthesia or sedation may also dictate the amount of injection sites [3,13].

## 4. Techniques for Localizing Muscles and Endplate Zones

After the introduction of pediatric BTXA treatment, MP, taking advantage of anatomical landmarks, the palpation of muscles, and joint passive movements, was the mostly used technique in both the upper and lower limbs [48,49,50]. However, researchers noted that palpation may be used with large, long, and superficially located muscles, such as GC, but is not typically sufficient with deep-seated, small, and slender muscles [15,51]. Therefore, manual approaches were usually combined with electrophysiological techniques, such as EMG or electrical stimulation (ES).

The drawbacks with these aforementioned techniques are the uncertainty of controlling the position of the needle tip and assuring the correct placement when finishing. Sedation may lower the EMG signal amplitude, and, on the other hand, continuous spastic muscle background activity may make it difficult to distinguish the MEP-noise. EMG does not always ascertain that the needle is in the right muscle. ES is better in verifying the right muscle but may need several replacements and is usually painful, requiring analgesia or sedation in children. Young children are rarely co-operative and dislike situations requiring them to stay still. Children with spastic CP also have limited control of specific muscles, which alters their ability to activate the muscles for EMG location.

Ultrasound has become a very helpful and widely used tool in pediatrics, offering a visually controlled option to potentate the accuracy of injections in children [52]. The advantages are less pain, the possibility to control the injection placement even with anatomic variations, the differentiation of neighboring structures (muscles, vessels, nerves, and bones), not requiring co-operation or sedation, and the choice of documenting the site and volume of the injected toxin [15,53]. However, ultrasound does not help in finding the MEPs.

In a study of 226 children with 502 muscles injected and a total of 1372 injections administered, the MP was checked with ES when children were injected under general anesthesia [51]. The accuracy of the placement was as follows: triceps surae 78%, hip adductors 67%, medial hamstrings 46%, tibialis posterior 11%, biceps brachii 62%, adductor pollicis 35%, pronator teres 22%, flexor carpii ulnaris 16% and flexor carpii radialis 13%. General anesthesia relaxes the muscles, which is likely to have impaired the ability to palpate the target muscles before ES. The authors concentrated on the accuracy of needle placement and utilized no functional outcomes. They recommended using ES for all muscles except the triceps surae.

In another study comprising 272 injections in GC muscles of 39 children with spastic CP, the MP was checked against ultrasound [54]. The accuracy was 93% into medial GC and 65% into lateral GC, and the accuracy was lower for children under four years of age (58%) than in the older age group (78%).

Xu et al. studied ES-guided and MP groups in their randomized controlled study of 65 di- and hemiplegic children [55]. The ES-guided patients received better functional gains in walking velocity and gross motor functions at three months post-injection compared to the MP group. In a small controlled trial comparing ultrasound and ES-guided GC muscle injections in 32 children with spastic equinus gait, no significant differences were discovered in muscle tone, dynamic muscle length, or selective motor control between the groups. However, the functional outcome in gait pattern significantly improved in the ultrasound group at one and three months post-injection [56].

In adult dystonic or spastic patients, the use of EMG-guidance helped to verify the right placement [57,58]. In spastic GC muscles after stroke, when comparing MP and ES and verifying the site with ultrasound, the accuracies were 88–93% and 92–95% for medial GC and 64–74% and 87–92% for lateral GC, respectively [59]. Neither MP nor ES showed full accuracy when checked against ultrasound. In spastic post-stroke forearm muscles, the overall accuracy of MP verified with ultrasound was 51%. The accuracy was higher for the finger flexors than for the wrist flexors (63% vs. 39%) [60].

Guidance with ultrasound appears to help to achieve an acceptable accuracy. None of the aforementioned studies attempted to determine whether the ES- or ultrasound guidance minimized side effects. Neither did they study whether the experience of the injector affects the rate of accuracy. In a cadaver study evaluating palpation in GC injections, the “experienced” (regular injector) and the “naïve” (little or no injection experience) groups were found to have success rates of 83% versus 17%, respectively [61].

## 5. Effect of Doses and Volumes in the Pediatric Setting

The standard expression of potency used for all BTXA-preparations is “mouse unit” (mu or U). One U is the amount of BTXA (in nanograms) that kills 50% of mice when injected intraperitoneally (the lethal dose 50, LD50) and each preparation contains different amounts of the 150-kD toxin per LD50 unit [62,63]. Due to the different bioavailabilities between the BTXA products, the recommended units are not equivalent. There is no agreed conversion ratio between onabotulinumtoxinA and abobotulinumtoxinA, but a large number of studies have reported a ratio of 1:3 [62,63]. The doses of incobotulinumtoxinA seem to parallel those of onabotulinumtoxinA and both the preclinical and clinical analyses have demonstrated a conversion ratio of 1:1 between onabotulinumtoxinA and incobotulinumtoxinA [63].

Doses, dilutions, volumes and diffusion (spread) are interrelated. In children, the dosage is given either in units per body weight (U/kg) or as the total dose in units. The size of the molecule complex seems to affect the fluid-based spreading of BTXA within the muscle (the larger the complex, the more limited the diffusion) which, in turn, shapes the side effect profile and safety margin [62]. The biological activity and adherence to the muscle tissue are stabilized by non-toxin associate proteins (hemagglutinins and non-hemagglutinins), which do not affect the efficacy. In animal and human models, the three BTXA products have been noted to demonstrate quite similar potency and diffusion profiles [62,63].

BTXA diffusion may be wanted or unwanted and may occur both within and outside of the injected muscle and into the adjacent muscles. When rat TA muscles were injected with increasing doses or volumes, doubling the area required a 25-fold increase in dose and a 100-fold increase in volume [35,64]. The dose effect was more potent than the volume effect when measured as reduction in muscle mass, post-injection dorsiflexion torque, and fiber cross-sectional area [64]. In rabbit latissimus dorsi muscle, a diffusion gradient over a distance of 3–4.5 cm in the injected muscle and over 1.5–2.5 cm into the contralateral muscle from the injection point was noted, crossing anatomic barriers, such as fascia and bones [65,66]. Although, there are various difficulties in directly detecting BTXA spread in the human muscle, based on the animal and human data available and on clinical experience. BTXA diffusion is likely to take place up to a distance of 4.5 to 5 cm and into adjacent muscles, depending on the dosage and volume injected [15,62,67].

Human studies suggest that an ideal dose per muscle exists, after which the effect subsides. Sloop et al. injected saline or varying doses of onabotulinumtoxinA (1.25 to 20 U/ 0.2 mL) into the EDB muscles in healthy adults and obtained a dose-dependent relationship with a logarithmic appearance between the dose and the decrease in the EMG M-response amplitude [20]. They showed a maximal denervation of 85–90% from 7.5 U onward, after which the curve reached a plateau. Additionally, a doubling of paralysis was obtained by a four-fold increase in dose up to 7.5 U. These findings were later replicated in adult cervical dystonia patient sternocleidomastoid muscle [68]. In children with spastic upper limbs, denervation up to 94% with an onabotulinumtoxinA dose of 1.4 U/kg was observed and the authors recommended injecting the forearm flexors with fewer than 1.5 U/kg per muscle [5]. The aforementioned studies did not report the incidence of side effects, but it is a widely held opinion that increasing the dose per site in a given muscle may saturate the NMJs, and increase the risk for the overflowing toxin to spread into the blood circulation and cause systemic side effects [13,14].

*Dosage.* Since the first report on pediatric BTXA treatment that used a total dosage of 1–2 U/kg, the amounts used at a single session have gradually increased as experience and research has extended. The amounts used in single muscles have not increased as much as the total dosage, which reflects the injecting more muscles per session as the multi-level treatment regime has become more widely adopted. The guidelines for treating children have been summarized based on the experience of experts in combination with the research known at the time of publishing [3,11,12,13,14]. In the European Consensus 2009, the maximum total amount per session was set at 400 (600) U or 20 (25) U/kg for onabotulinumtoxinA and 1000 U or 20 (30) U/kg for abobotulinumtoxinA [12]. The clinicians are recommended to be cautious and to refer to the product information for individualization of the dose. The base for dose calculations and the dose modifiers are depicted in Table 1. Adult dosing recommendations are to be substituted for children heavier than 60 kg [11].

In clinical studies, the doses in the lower limbs have ranged from 1 to 12 U/kg of onabotulinumtoxinA or 15 to 30 U/kg of abobotulinumtoxinA for the gastroc-soleus muscles and 1 to 5 U/kg of onabotulinumtoxinA or 20 to 30 U/kg of abobotulinumtoxinA for the hamstrings and adductors [16,69,70]. Randomized controlled studies focusing on the effects of different doses in children are few and most of them are confined to the calf muscles. abobotulinumtoxinA at 10 U/kg to 15 U/kg of per leg was found to be appropriate doses for the gastroc-soleus muscle [71,72,73].

In a clinical cohort comprising 758 patients with 1594 single- or multilevel treatments Bakheit et al. noted that children receiving a total dose between 10 and 40 U/kg of abobotulinumtoxinA benefited the most in terms of reaching their goals [74]. The multi-level injections of smaller doses into sole muscles were found to be safer and more efficient than single-level high doses. Two cohort studies on children with CP found that the use of doses higher than 6 U/kg in the gastroc-soleus did not produce better or longer lasting effects compared with lower doses, and the best result in gait pattern was gained in the diplegics with the dose of 3–4 U/kg and in the hemiplegics with a dose of 5–6 U/kg of onabotulinumtoxinA [75,76].

Doses of 0.5–2 U/kg for the forearm muscles, 2–3 U/kg for the arm muscles, and a total of 5–7.5 (10) U for the adductor or opponens pollicis, have been recommended [3,11]. In clinical studies, the doses for each muscle demonstrated a wide range of variation: 0.3–4 U/kg for the arms, 0.5–4.9 U/kg for the forearms, and 0.9–1.8 U/kg for the adductor pollicis with onabotulinumtoxinA [14,16,77]. In a double-blind randomized upper limb trial, the doses in the higher dosage group were double the doses in the lower dosage group [78]. The study showed no differences between treatment groups in function, ROM, muscle tone, or grip strength. In order to keep the injector blinded, the authors used varied dilutions of 50–200U/mL onabotulinumtoxinA, which may have inclined the results in favor towards the low-dose group. As their conclusion, the following doses were recommended: Biceps brachii 1 U/kg, wrist/finger flexors 1.5 U/kg, brachioradialis and pronator teres 0.75 U/kg, and adductor/opponens pollicis 0.3 U/kg (total max. dose 10 U). The suggested doses are summarized in Table 2.

*Dilution.* The most common dilutions used in pediatric studies have been 100 U/1 mL or 100U/2 mL (range 1−4 mL saline/vial) for onabotulinumtoxinA and 500 U/2.5 mL (range 1−5 mL saline/vial) for abobotulinumtoxinA [12,13,14]. The impact of the dilution on the rate of spread is not known. Animal studies indicated that increasing the injection volume increases muscle paralysis as well. However, how this translates into clinical practice is still under debate [35,64,79]. In humans, conflicting results have been attained as some have gained no differences between the groups (dilution ratio of 1:2 or 1:4) as others have shown significant improvement in the muscle tone and electrophysiological measurements (dilution ratio of 1:5) [42,80,81,82,83].

## 6. Recovery of Nerve Endings and Muscles

BTXA does not cause cell death or axonal degeneration, and the nerve stays in contact with the muscle. The human NMJs recover by developing sprouts from the end plate, the preterminal axon, and adjacent nodes of Ranvier [84]. In an in vivo mouse muscle model, the new sprouts were able to activate the muscle four weeks post-injection, after which the sprouts degenerated, and the original terminal axon regained its function in three months with normal Ach-release and the synthesis of new SNARE protein [85]. Apparently, the recovery of NMJs can take place many times without loss or destruction of the NMJ function, and this is the rationale for repeated BTXA injections in human therapy. The findings of the de Paiva study led to the recommendation of having at least three months of time before repeating the BTXA treatment in children [3]. Unless toxin neutralizing antibodies or fixed muscle contractures intervene, the muscles are thought to continue to respond after multiple injections [84].

Skeletal muscles contain a specific mixture of type I slow, type IIa fast fatigable, and type IIb fast fatigue resistant fibers, and each fiber type expresses slow or fast myosin heavy chains, with differing metabolic and force production properties. In rodents, a shift of fiber phenotype from fast to slow was detected, as estimated by an increased expression of type I and IIa myosin heavy chain in BTXA-treated rat muscles [86,87,88,89]. Whether this shift reverses in time is still unclear, but a higher expression of slow type isoforms was observed even at 18 months after treatment [89]. In addition, an accumulation of collagen, decrease in fiber size and cross-sectional area, reduction in fiber stiffness and slack sarcomere length, increase in fiber bundle passive elastic modulus. Numerous transcriptional adaptations, activating the gene pathways of repair and atrophy, were noted [87,88,89,90,91].

The treatments affect the muscle force, mass, and contractile properties as well. In rodent and rabbit models, after a single BTXA injection, a loss of muscle force by 50−90%, of muscle mass by 43−80%, and of histological contractile material by 4% was noted [86,89,92]. After three and six once in a month given injections, the muscle force was reduced by 89% and 95%, the muscle mass by 45% and 30%, and the muscle contractile material by 19%, and 36%, respectively [92]. After one month onward and in the contractile material from three months onward. At the end point after six months of recovery, the values did not quite reach those of the control group [93]. Electrical stimulation exercise three times per week on the injected muscles during the whole six month, once in a month treatment period partially mitigated the degeneration [94].

Notably, there is still no animal model that accurately mimics the changes observed in human spastic muscles and the studies were performed with normal muscles. The BTXA doses used in these animal studies were relatively high. Whether the changes in the treated or non-treated muscle were due to the treatment itself or from not using the limb due to weakness was also unclear.

In human orbicularis oculi muscles, a total recovery was detected six months after injection and no persistent histological changes were detected even after several years of multiple blepharospasm treatments [84,95,96]. In two healthy adults, significant neurogenic muscle atrophy was detected on MRI (14−19% reduction at 3 months; 27% at 6 months; 13−38% at nine months; 12−22% at 12 months) confined to the injected lateral GC muscle after a single injection (75 U of incobotulinumtoxinA) [97]. The histopathology revealed neurogenic atrophy of tiny groups of muscle fibers, with some compensatory hypertrophic fibers, minor local fiber type grouping, frequent target fibers showing re-innervation, some accumulation of endomysial connective tissue and a mild increase of perimysial fat cells one year following treatment. No changes in the MRI or histology were detected in the non-treated contralateral muscle. Therefore, the treated muscles did not quite gain their initial size and high-signal intensity pattern within the one year observation period.

## 7. What About the Muscles of Children with CP?

In CP, the brain lesion leads to complex aberrant neuronal input in ways that are not yet fully understood. The non-progressive central nervous system (CNS) lesion results in progressive secondary musculoskeletal pathology. Spasticity, a velocity-dependent resistance to stretching occurs during the first developmental years, due to the reduced inhibition of the stretch reflexes. The impaired muscle growth, and contractures (i.e., fixed muscle shortening) restricting the range of joint movement developed thereafter [98,99,100]. It is important to distinguish between the neural CNS-related and the non-neural tissue-related stiffness of passive resistance when assessing spasticity [99]. In addition, stiffness may arise from other structures, such as joint capsules, tendons, fascias, ligaments, or skin.

The progression of dynamic spasticity to fixed contractures is the fundamental determinant for effective BTXA treatment. The majority of children with CP may have a combination of both [2]. In addition to spasticity, other impairments, such as co-contraction, impaired selective motor control, muscle weakness and sensory deficits, may play a role in spastic CP. The mechanisms of contracture formation and impaired muscle growth are not yet fully known. It is widely held that the reduction of muscle growth appears first, and the joint stiffness follows some years later, and that the discrepancy between bony and muscle growth is responsible for the secondary increase in muscle stiffness [99,100]. The extent of the brain lesion, the distribution and degree of motor impairment (Gross Motor Function Classification Systems level, GMFCS), the degree of activity, and the treatments used (physiotherapy, orthotics, casting, strength training, surgery, and BTXA) are expected to further influence the muscle properties [101].

BTXA injection reduces the spasticity and increases the joint range but does not appear to decrease the passive intrinsic muscle stiffness or affect muscle compliance [102,103,104]. The first or subsequent injection, administered into the GC, was shown to increase the muscle length for a period of 12 to 24 weeks, as measured with three-dimensional gait analysis [105]. In ultrasound studies, the GC fascicle angle was reported to decrease at the ankle resting, neutral, or maximum dorsiflexion position at 1 and 3 month post-injection, and the fascicle length was reported to increase at resting and maximum dorsiflexion [106,107]. In a histopathological study comparing repeatedly BTXA-treated GC and non-treated vastus lateralis biopsies, neurogenic atrophy was seen in 60% of the participants between four months and three years post-treatment [108]. Fiber type 1 loss and type 2 predominance were also observed, which is contrary to previous animal and human results [108]. In preliminary gene expression studies, when the treatment effect was examined by comparing injected upper limb muscles versus non-injected muscles, no altered transcripts of statistical significance were detected [109].

BTXA causes muscle wasting that is physiologically consistent with immobilization or surgical denervation or recession [110,111]. In ultrasound and MRI studies, muscle atrophy was detected for between 5% and 20% of the injected muscles (GC, medial hamstrings, or iliopsoas) at 1 to 3 months, and for 7% at six months [38,112,113]. The loss of volume peaked between 1 and 3 months, after which the muscle started to recover, and the original volume was fully or almost fully achieved in one year [113,114,115]. The first exposure to BTXA seemed to result in greater atrophy compared with subsequent exposures [113,115]. The differences in muscle volume, fascicle length, or cross-sectional area were not significant between the single and multiple (three injections in one year) injection groups [115]. Both the GMFCS level, and the history of more than three BTXA injections were associated with increased echo-intensity and decreased medial GC volume, reflecting structural alterations in the muscles [116].

The aforementioned studies reported no atrophy in the adjacent and antagonistic muscles nor deterioration in the function and strength. On the contrary, compensatory hypertrophy was observed in the soleus or hamstring muscles when the GC muscles were treated. The slight increase in isometric plantar flexor strength was attributed to the children’s ease of achieving the range of motion after treatment and the compensatory synergistic muscle hypertrophy [112]. The majority of these studies were performed on the GC muscles of children with GMFCS levels of I and II, aged 5−16 years with hemi- or diplegia. The doses were reported to be between 1.4−6 U/kg of onabotulinumtoxinA using 1 to 2 mL dilutions. The possible dose-response with muscle volume loss was not explored. However, it is reassuring that the extent of atrophy appeared to be much smaller than in the animal studies.

Muscle weakness is recognized as a remarkable impairment and, for children already predisposed to deficits in muscle size and strength, a treatment that potentially leads to further weakening is a concern. Recently, more studies have come out exploring the benefits of individually tailored resistance training for muscle strength and there is preliminary evidence that strength training increases the cross-sectional area, volume, and thickness of the muscles [117,118,119,120]. Combined strength training and BTXA treatment has been reported to improve outcomes over BTXA treatment alone with no muscle atrophy noticed after injection [121,122,123]. The MRI results at six months of follow-up showed reduced spasticity with improved muscle strength associated with muscle hypertrophy [123]. All children showed strength gains regardless of the timing of the training period. The authors postulated that the pre-treatment training could be more suitable for a child scheduled to have serial casting following BTXA, whereas post-treatment training could be more appropriate if immediate functional gains were the goal.

## 8. Lessons Learned from the Repeated Treatment Studies

Initially, regular injections were thought to delay or reduce the formation of fixed muscle contraction and the need for surgical interventions [105]. Such assumptions were based on discoveries in a “hereditary spastic mouse model”, where the toxin being injected into the GC of baby mice before they developed spasticity, prevented the development of muscle contracture [124]. However, the maturing and growth of a human takes a much longer time compared with the mouse, and the properties of muscles between mice and humans differ, making the comparison unjustified.

In a systematic review including four upper extremity and nine lower extremity studies investigating the effects of repeat BTXA, the authors postulated positive effects on spasticity, ROM, gross motor function, gait, and hand function [125]. The injections were repeated 1−13 times and the intervals varied from 3 to 12 months. The adverse events were minor, transient, and few in number. Interestingly, the first two injections provided more improvement than the later ones. The number of children who were followed up decreased as the number of injections increased, possibly due to enhanced contractures and improvements decelerating with increasing age. The authors concluded that repeated injections were effective, but that, in general, distinguishing the BTXA-caused functional improvements from natural growth or any co-intervention was difficult.

With respect to the preferred spastic equinus treatment interval, two independent studies with over two years follow-up found no significant differences between the “every 4 months” or “every 12 months” injected groups in passive ankle ROM, motor function, or gait [126,127]. In the subgroup of hemiplegics, the passive ankle ROM decreased by a mean of 8.5 degrees in two years’ time, and even tripling the frequency of injections did not prevent this progression. In the diplegics’ groups, the passive ROM increased on average by 1.6 degrees, more so in the 4-month group, but at the cost of concomitantly increasing crouch gait [127]. The authors recommended 12 month intervals between BTXA treatments in spastic equinus gait.

The effect of BTXA injections on ROM are commonly reported to be of short duration, despite the reduction in spasticity. In a prospective cohort study of 94 children, with different CP-subtypes and GMFCS levels I−V, the children received 1−8 repeated injections at a minimum of six month intervals into the GC, hamstring, and adductor muscles [128]. Assessments of the muscle tone and ROM were made before and three months after each treatment, with the longest follow-up at three years seven months. An immediate reduction in the muscle tone was observed after each injection. However, the improvement in ROM was short-lived and increased only after the first injection, after which the ROM declined. The authors concluded that repeated BTXA injections did not prevent the development of fixed contractures, and that the development of contractures was not associated with increased muscle tone [128]. At this point, the treatments should be discontinued and the patients referred to surgery when applicable [2].

Long-term results after repeated BTXA injections and follow-up are emerging as the children who were first treated are now reaching their adulthood. Population-based health programs and CP-registers are a possible source of information to shed some light on these issues. There is some evidence to propose that, combined with other treatment modalities, the injections may have helped to maintain or improve function at a later age, postponed the age for surgery, reduced the incidence of re-operations, and lowered the prevalence of surgical procedures, e.g., Achilles tendon lengthening or multi-level interventions [129,130,131,132].

Understandably, improvements over a long time period are mixed with natural development and a combination of other treatment modalities. According to a recent systematic review, single or repeated BTXA treatments, either alone or combined with bracing, appeared to decelerate but not to prevent hip displacement [133]. Orthopedic surgery is still important for the prevention and reconstruction of hip dislocation, and to correct and stabilize scoliosis and foot deformities [131].

## 9. Conclusions

In busy clinics, the calf muscles are the most common targets for BTXA injections and, as they are superficially situated, they are often thought to be easy to inject without technical guidance. However, in children with CP under the age of eight years, the triceps surae muscles are surprisingly thin: medial GC 0.9 cm, lateral GC 0.75 cm, and soleus 0.98 cm, on average [54,56]. The use of ES, and particularly of ultrasound, increased the accuracy of localizing the right target muscles and the depth of the needle tip, more so with the slender, thinner, or deeper muscles and children with more subcutis [134].

Animal studies suggested that BTXA spreads up to 4.5−5 cm from the injection site. Concluding from the spastic equinus studies, BTXA appears to spread efficiently within the muscle and has a high affinity to the NMJs. Therefore, injecting the muscle may suffice to reduce the muscle tone. Nevertheless, targeting close to the NMJs is preferred. As knowledge of the sites of NMJs in different upper and lower limb muscles, with different muscle patterns, has increased, the placement of the needle with better probability is possible. Further studies with the larger and differently configured lower limb muscles may be required. In the upper limbs, BTXA presumably diffuses sufficiently throughout the muscles, and this issue warrants further studies as well. The available outcome measures may not be sensitive enough to differentiate the functional effect between groups, if there is one.

The present dose studies indicated that increasing the dose over 6 U/kg of onabotulinumtoxinA or 15−20U/kg of abobotulinumtoxinA in the calf muscle, or the onabotulinumtoxinA doses per muscle in the upper limb over 1.5−2 U/kg in the arm, over 1−1.5 U/kg in the forearm, and over 10 U in the adductor pollicis muscle did not lead to better results in reducing tone and increasing function. Patients responding poorly to repeated injections of adequate doses should be re-evaluated for fixed muscle contractures and treated accordingly. Clinicians should pay attention to the non-linear dose response to BTXA and track both the intended and adverse effects of their treatment practice. The incidence of adverse events increases with higher total doses and cautiousness, in particular for those children with more severe disabilities, is necessary [74,135]. If increasing the doses gives no advantage, volume changes might be an option. Studies with different dilutions have, thus far, yielded inconclusive results. Further trials with constant doses, but various volumes with adequate dilution ratios are warranted.

The extent of post-BTXA treatment muscle atrophy in children with CP appears to be smaller than the animal studies would suggest. The doses and the treatment intervals (smaller doses, longer intervals between treatments, and selecting different muscles or injection sites within the muscle) are to be weighed against the whole rehabilitation plan and other adjunctive treatments. Whether repeated BTXA sessions further inhibit muscle growth in CP remains unclear. However, without non-treated control groups it is not possible to know whether this effect is due to the injections, the reduced growth in CP in general, or other factors. Clearly, more longitudinal studies with controlled investigations are needed. Additionally, the role of serial casting in possibly further increasing muscle atrophy has not been explored.

The encouraging results of home-based upper and lower limb strength exercises combined with BTXA treatment could be implemented into CP practice. Strength training of both treated and antagonist muscles could be utilized. The training session length of 40−50 min three to five times per week lasting for a minimum of 6−8 weeks showed the largest effect size. Although, two to three times per week may be the most feasible [119]. Strength training may be, to some extent, incorporated as part of the children’s standard physiotherapy or every-day care. Currently, this training could be supervised via Internet connections along with the traditional sessions, considering the developmental age of the child, the family situation, and the day-care/school environment. How long the effect lasts and how long to practice are not yet defined, and motivation in the teen population may also become a challenge.

## Figures and Tables

**Figure 1 toxins-12-00440-f001:**
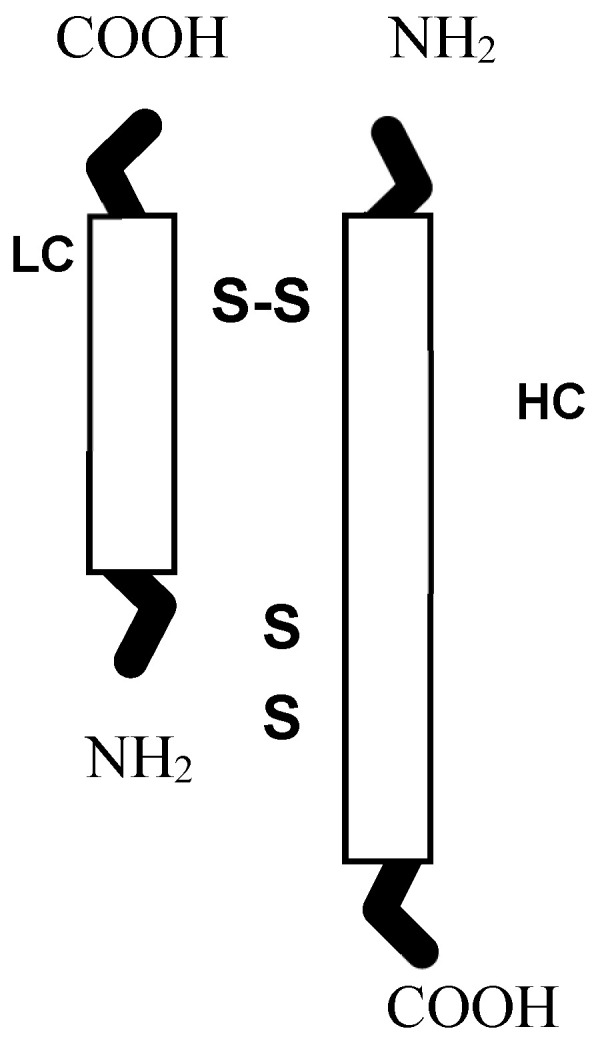
Structure of botulinum toxin type A (BTXA). LC = light chain, HC = heavy chain.

**Figure 2 toxins-12-00440-f002:**
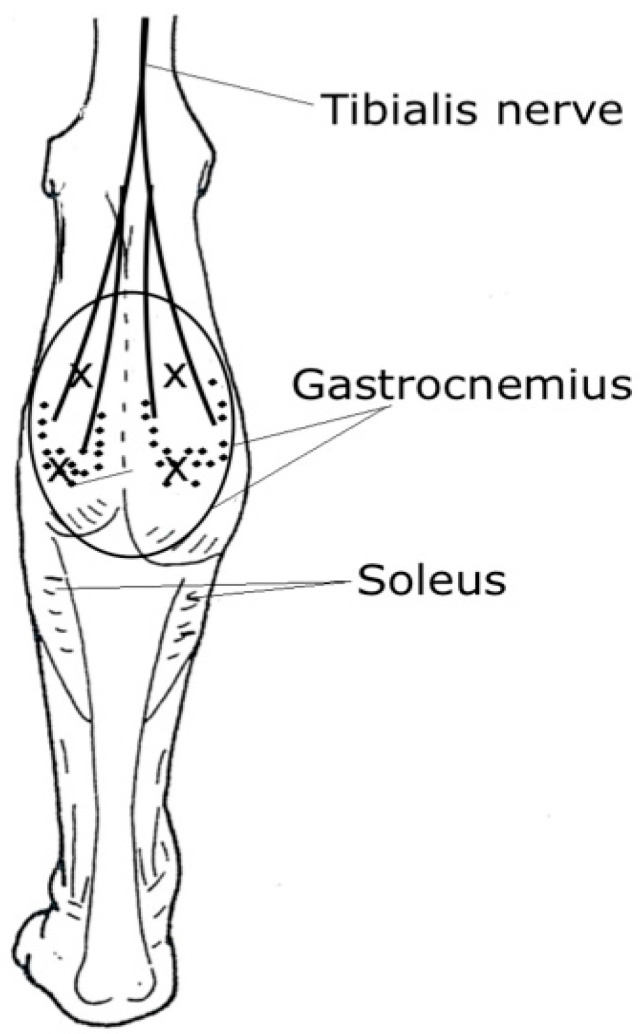
An illustration of the motor endplates (MEP) (dots), injection sites (x), and the area of diffusion along the gastrocnemius muscle (large circle). According to Parratte et al. [29], the MEPs are distributed in zones confined mostly to the mid-part of the muscle belly.

**Table 1 toxins-12-00440-t001:** The base for dose calculation and dose modifiers [12].

**Base for Dose Calculation:**
Total units per treatment session
Total units per kg body weight per session
Units per muscle
Units per injection site
Units per kg body weight per muscle (U/kg/muscle)
**Dose Modifiers:**
Weight of the patient
Number of muscles needing treatment
Size and activity of the muscle
Knowledge of the muscle configuration and MEP distribution
Severity of spasticity/dystonia
The goal of the treatment
Comorbidities (e.g., dysphagia, aspiration, and breathing problems)
Dynamic vs fibrotic muscle
Experience from previous injections

**Table 2 toxins-12-00440-t002:** The recommended dose ranges for muscles summarized from [3,5,6,11,14,71,72,73,75,76,78].

Muscle	Dose Range U/kg of OnabotulinumtoxinA		Dose Range U/kg of AbobotulinumtoxinA	
Gastrocnemius mediale	1–3		3–6	
Gastrocnemius laterale	1–3	total max 4–6 U/kg	3–6	total max 10–15 U/kg
Soleus	1–2		2–4	
Tibialis posterior	1–2		--	
Semitendinosus	1–3		10–15	
Semimembranosus	1–3		10–15	
Biceps femoris	1–3		--	
Adductor longus/magnus	1–4		20–30	
Gracilis	1–2		--	
Flexor carpi radialis	0.5–1.5(2)		5–10	
Flexor carpi ulnaris	0.5–1.5(2)		5–10	
Pronator teres	0.75–1(2)		5–10	
Brachioradialis	0.75–1(2)		5–10	
Flexor digitorum superficialis ^1^	(0.5)1–1.5(2)		5–10	
Flexor digitorum profundus ^1^	(0.5)1–1.5(2)		5–10	
Biceps brachii/Brachialis	1–2 (2–3)		5–10	
Adductor pollicis ^1^	0.3–1; Total max 10–15 U		3–5; total max 45 (−75) U	
Flexor pollicis brevis ^1^	0.3–1; Total max 10–15 U		3–5; total max 45 (−75) U	

^1^ The dose used depends on the planned goal of the thumb/fingers contributing to the grasp or not.

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
