# Peer review of "Over 25 Years of Pediatric Botulinum Toxin Treatments: What Have We Learned from Injection Techniques, Doses, Dilutions, and Recovery of Repeated Injections?"

_toxins, 2020, doi:10.3390/toxins12070440_

Round 1

Reviewer 1 Report

This is a very interesting review paper showing clinical usage of botulinum toxin type A. I have three issues which should be corrected before publication:

In paragraph 2 should be provided information about botulinum toxin toxic action and potentially dangerous of using it

Figure 1 should be corrected; some places are cut

The described used dosage of botulin toxin should be compared with its LD50 parameter

Author Response

Thank you very much for the reviewers for their constructive suggestions. We have made the following revisions.

Rev 1.

This is a very interesting review paper showing clinical usage of botulinum toxin type A. I have three issues which should be corrected before publication:

  1. In paragraph 2 should be provided information about botulinum toxin toxic action and potentially dangerous of using it.

The information about toxin action and potential side effects were added into paragraphs 3 and 4:

“The toxin is produced by the anaerobic spore-forming bacteria Clostridium botulinum of which seven immunologically distinct serotypes designated with alphabetical letters from A to G have been identified. They exert their effect by blocking the release of the neurotransmitter acetylcholine at cholinergic nerve endings of the skeletal and autonomic nervous system. A selective and temporary chemical denervation ensues, causing clinically detectable muscle weakness and atrophy.”

“The adverse events reported have usually been transient and mild, such as focal weakness, pain at the injection site, bruising, tripping, a local rash, and influenza-like illness. Fewer reports have included symptoms such as urinary or fecal incontinence, generalized weakness, worsening of strabismus or dysphagia, irritability, or constipation. Few deaths have been reported in the literature, and caution is recommended in children with pre-existing bulbar symptoms, gastro-esophageal reflux, or frequent chest infections, as these conditions expose patients to aspiration pneumonia [3]. Other contraindications include myasthenia gravis and the concomitant use of amino glycoside antibiotics or non-depolarizing muscle relaxants.”

  1. Figure 1 should be corrected; some places are cut.

The figure 1 has been corrected.

  1. The described used dosage of botulin toxin should be compared with its LD50 parameter

The information about toxin LD50 parameter was added into paragraph “Effect of doses and volumes in the pediatric setting”:

“The standard expression of potency used for all BTXA-preparations is “mouse unit” (mu or U). One U is the amount of BTXA (in nanograms) that kills 50% of mice when injected intraperitoneally (the lethal dose 50, LD50) and each preparation contains different amounts of the 150-kD toxin per LD50 unit [62,63]. Due to the different bioavailabilities between the BTXA products, the recommended units are not equivalent. There is no agreed conversion ratio between onabotulinumtoxinA and abobotulinumtoxinA, but a large number of studies have reported a ratio of 1:3 [62,63]. The doses of incobotulinumtoxinA seem to parallel those of onabotulinumtoxinA and both the preclinical and clinical analyses have demonstrated a conversion ratio of 1:1 between onabotulinumtoxinA and incobotulinumtoxinA [63].

Reviewer 2 Report

The authors have written a thorough review on the injection techniques, doses and dilutions, the recovery of muscles from BTXA injections and how it impacts pediatric treatments. I have few suggestions for improvement.

  1. Line 53 Accurate injection directed as close as possible to
     the neuromuscular junctions (NMJ), has been considered to be a prerequisite for efficient treatment.  In the introduction the importance of accurate injections is mentioned. Would be best to have a figure showing the NMJ and the target location (MEP) to have a better visualization for this review. Also illustrating how much the BTXA can diffuse through the muscles.
  2. The figure of the structure of BTXA has unclear labelling. Needs to be fixed. 
  3. A table summarizing the appropriate doses would be helpful for the review.

Author Response

Thank you very much for the reviewers for their constructive suggestions. We have made the following revisions.

Rev 2.

The authors have written a thorough review on the injection techniques, doses and dilutions, the recovery of muscles from BTXA injections and how it impacts pediatric treatments. I have few suggestions for improvement.

  1. Line 53 Accurate injection directed as close as possible to
     the neuromuscular junctions (NMJ), has been considered to be a prerequisite for efficient treatment.  In the introduction the importance of accurate injections is mentioned. Would be best to have a figure showing the NMJ and the target location (MEP) to have a better visualization for this review. Also illustrating how much the BTXA can diffuse through the muscles.

Figure 2 was created to illustrate the MEP zone, site of injection and diffusion area using gastrocnemius muscle as an example (please, see the manuscript). According to Parratte et al (ref 29) the MEPs are distributed in zones confined mostly to the mid-part of the muscle belly.

  1. The figure of the structure of BTXA has unclear labelling. Needs to be fixed. 

Figure 1 has been fixed.

  1. A table summarizing the appropriate doses would be helpful for the review.

We added a summary table (Table 2) to give a more holistic picture of the recommended doses. We picked the doses from the references used in the review article. Please, see Table 2 in the paragraph “Effect of doses and volumes in the pediatric setting”.

Reviewer 3 Report

This article provides a review of studies on BTXA treatment: injection techniques, doses, dilutions, and the impact of repeated injections.  It is well organized and informative. 

One minor suggestion is to provide a model figure on the motor endplate zone at the NMJs.  It will help readers to understand the paper.

Another minor suggestion is to spell ultrasound instead of using "US".  

Author Response

Thank you very much for the reviewers for their constructive suggestions. We have made the following revisions.

Rev 3. Comments and Suggestions for Authors

This article provides a review of studies on BTXA treatment: injection techniques, doses, dilutions, and the impact of repeated injections.  It is well organized and informative. 

  1. One minor suggestion is to provide a model figure on the motor endplate zone at the NMJs.  It will help readers to understand the paper.

Figure 2 was created to illustrate the MEP zone, site of injection and diffusion area using gastrocnemius muscle as an example (please, see the manuscript). According to Parratte et al (ref 29) the MEPs are distributed in zones confined mostly to the mid-part of the muscle belly.

  1. Another minor suggestion is to spell ultrasound instead of using "US".  

                All the abbreviations “US” were changed to “ultrasound”.